

# Structural and magnetic anisotropy in $YBa_2Cu_3O_7$/$La_{0.67}Sr_{0.33}MnO_3$ bilayer on $SrTiO_3$ substrate

**Ankita Singh, Ram P. Pandeya, Sawani Dutta, Srinivas C. Kandukuri and Kalobaran Maiti**$^\star$

Department of Condensed Matter Physics and Material Science,
Tata Institute of Fundamental Research, Homi Bhabha Road,
Colaba, Mumbai - 400005, India

$\star$ kbmaiti@tifr.res.in

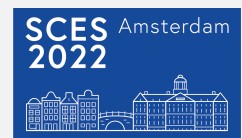

*International Conference on Strongly Correlated Electron Systems
(SCES 2022)
Amsterdam, 24-29 July 2022*

## Abstract

We study the magnetic properties and emergence of superconductivity in $YBa_2Cu_3O_7$ (YBCO)/ $La_{0.67}Sr_{0.33}MnO_3$ (LSMO) heterostructures. Bilayer films of superconducting layer, YBCO, and ferromagnetic layer, LSMO were grown on $SrTiO_3$ (STO) (001) substrate using ultrahigh vacuum Pulsed Laser Deposition system. Magnetization data at 100 K as a function of applied magnetic field shows ferromagnetic behaviour due to the LSMO layer. Cooling below 100 K leads to superconductivity in this material; the onset of superconductivity occurs at a temperature, $T_{c,SC}$, of 86 K for $H_{ext} \perp c$ (in-plane) & 84 K for $H_{ext} \| c$ (out-of-plane) under 100 Oe applied field. In-plane magnetic measurements show significant suppression of diamagnetic behaviour as compared to the out-of-plane measurements. The susceptibility signals are higher for out-of-plane direction. Such strong anisotropy in magnetism and the transition temperatures reveal complex interplay of magnetism and superconductivity in this system and calls for further study in this direction.

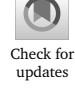
## 1 Introduction

Coexistence of superconductivity and ferromagnetism are barely been found in the same bulk material despite numerous efforts in this direction. It is realized that heterostructures in the form of thin films can allow to study the coexistence of superconductivity (SC) and ferromagnetism (FM) in the same material [1, 2]. This has opened up gateways to study the interplay of spins in the superconductivity region due to the presence of ferromagnetic interface. The appearance of mutual interaction between two different and competing ordering parameter

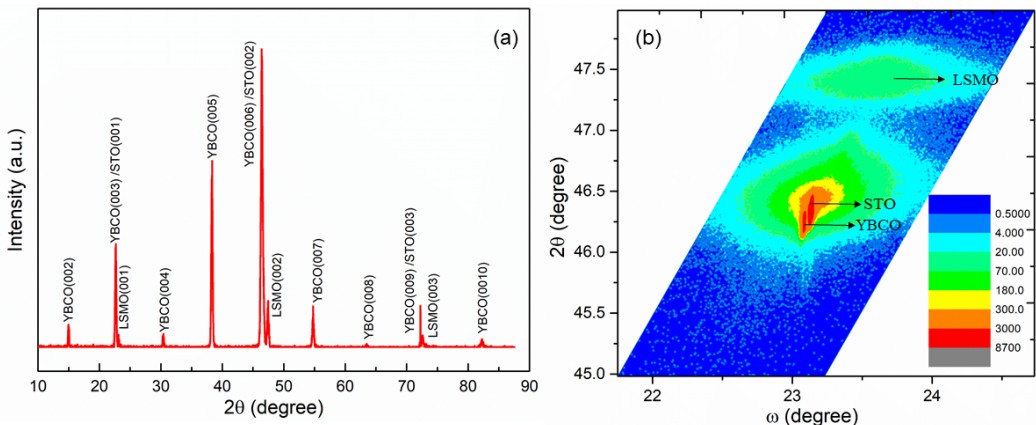

Figure 1: (a)X-ray diffraction pattern for YBCO/LSMO/STO bilayer thin film in 10° -90° range showing the uniaxial growth of both the layers along $c$ direction, (b) Reciprocal space mapping for the bilayer film along (002) direction of STO showing YBCO(006) as well as LSMO(002) layer.

leads to variety of novel phenomena like spin-triplet superconductivity [3, 4], domain-wall superconductivity [5], superconducting spintronic [6], etc. Exploiting the flexibilities in heterostructures, we investigate the properties of the heterostructure of ferromagnetic and superconducting layers employing magnetic measurements.

In the past, different exotic effects are observed due to the interface reconstruction of two different kinds of perovskite oxides like the appearance of 2D ferromagnetism at the interface [7], presence of 2D electron gas [8], alteration of preferred magnetic direction [9], etc. For example, Gibert $et.al.$ studied the heterostructure for $LaNiO_3/LaMnO_3$. The Mn to Ni charge transfer leads the development of interface induced magnetism in paramagnetic $LaNiO_3$ layer. First principle calculations indicate that this interfacial interaction gives rise to unusual spin ordering that resembles spin density waves in $LaNiO_3$ layer [10]. In another study heterostructure with different structured oxides show interesting properties. Zhang $et.al.$ showed the appearance of perpendicular magnetic anisotropy by inserting a $La_{0.66}Sr_{0.34}MnO_3$ layer between two $LaCoO_{2.5+\delta}$ layers. In such heterostructures, there occurs an interfacial orbital reconstruction due to lattice mismatch that resulted in an elongation of $MnO_6$ octahedron [11].

In this work, we have fabricated heterostructures of $La_{0.67}Sr_{0.33}MnO_3$ (LSMO) and $YBa_2Cu_3O_7$ (YBCO) on $SrTiO_3$ (STO) substrate using ultrahigh vacuum pulsed laser deposition system. We have performed systematic investigation to study the effect of FM/SC interface on the occurrence of perpendicular magnetic anisotropy in the heterostructure compound.

## 2 Experimental Details

Epitaxial and single crystalline thin films of YBCO/LSMO were deposited on single crystalline STO (001) substrate using a home built ultrahigh vacuum (UHV) pulsed laser deposition (PLD). For the deposition we have used KrF excimer laser operating at a wavelength of 248 nm. The deposition parameters for LSMO and YBCO were optimized individually and pulse repetition rate was maintained at 5 Hz. On STO substrate, initially a thin layer of LSMO was deposited. For the deposition of LSMO layer, substrate temperature was maintained at 750°C. The LSMO target was ablated for 3000 pulses that resulted into a thin film of thickness 20 nm. Oxygen partial pressure of the UHV chamber was brought up using high purity oxygen; the pressure was maintained at 300 mTorr during the LSMO layer deposition. On the top of

LSMO layer, YBCO layer was deposited. During the deposition of YBCO layer, substrate temperature was increased and maintained at 800°C. YBCO target was ablated for 21,000 pulses to deposit a thin film of thickness 50 nm. During the YBCO deposition, the chamber pressure with oxygen was maintained at 400 mT. After the deposition, the substrate was cooled down to 550 °C and annealed in the abundant oxygen pressure for 2 hours. After annealing, substrate was cooled down slowly to room temperature.

To identify the structural phase, epitaxial and single crystallinity nature of the deposited thin film, $x$-Ray diffraction (XRD) was performed on Bruker D8 X-ray Diffractometer. DC Magnetization measurements were performed on superconducting quantum interference device (SQUID)- vibrating sample magnetometer (VSM) of Quantum Design (QD) in the temperature range 2-300 K under an applied field of 100 Oe. Magnetic hysteresis loops of the thin films were measured at different temperatures (20 K & 100 K) under an applied magnetic field up to ± 6T.

## 3   Results and Discussion

Pseudo cubic lattice parameter for STO substrate is reported to be 3.905 Å [12]. Pseudo cubic lattice parameter of LSMO ($a \sim 3.876$) [12] and orthorhombic lattice parameter of YBCO ($a = 3.82$, $b = 3.89$) [13] is comparable with the lattice parameter of STO. In Fig. 1(a), we show the out-of-plane $\theta$-$2\theta$ measurement for the bilayer thin film. The XRD pattern shows only (00$l$) Bragg peak for both LSMO and YBCO layers indicating the heterostructure growth of the thin film along $c$-direction. The growth of comparatively thick YBCO layer did not degrade the growth of LSMO layer. In Fig. 1(a), XRD pattern shows the presence of peak from the substrate STO with the LSMO peak present on the right hand side of the substrate peak indicating lower inter-planar spacing for film in comparison to the substrate. Fig. 1(b) shows the reciprocal space mapping of (002) symmetric peak of STO with (002) and (006) peak of LSMO and YBCO respectively. These results suggest single phase of the samples grown with high degree of single crystallinity.

Magnetization versus temperature measurements were performed in the temperature range 2-300 K under an applied magnetic field of 100 Oe along two directions i.e. $H_{ext} \perp c$ (in-plane) and $H_{ext} || c$ (out-of-plane) direction showing a strong relationship between magnetic anisotropy and uniaxial growth of the thin film . Fig. 2 shows the magnetization Zero Field Cooled (ZFC) and Field cooled (FC) data along both the directions. The sample was cooled down to 2 K under zero external field and applied field for ZFC and FC case, respectively, and the measurements were performed during the warming of sample under an applied magnetic field of 100 Oe. Both ZFC and FC curves almost coincide down to 86 K (in-plane) and 84 K (out-of-plane) and then reduces sharply indicating the onset of superconductivity. Such dependence of the onset temperature for superconductivity, $T_{c,SC}$ on the direction of externally applied magnetic field is confirmed from the first derivative of susceptibility along both the direction (see the inset in Fig.2), reveals the anisotropy in the properties of the bilayer film. The Curie temperature for LSMO thin film is reported to be around 360 K [14]. In the case of our bilayer thin film, Curie temperature for LSMO is found to be higher than 300 K. The ZFC curve shows a broad peak in the superconducting transition region whereas the FC curve shows a gradual decrease with warming for the in-plane direction while almost constant behaviour in out-of-plane direction. For the 100 Oe applied field, the ZFC-FC susceptibility values are higher when external magnetic field is applied in the direction parallel to the $c$-direction (out-of-plane direction of the $CuO_2$ superconducting plane) as compared to the magnetic field applied in perpendicular $c$-direction (in-plane) [15].

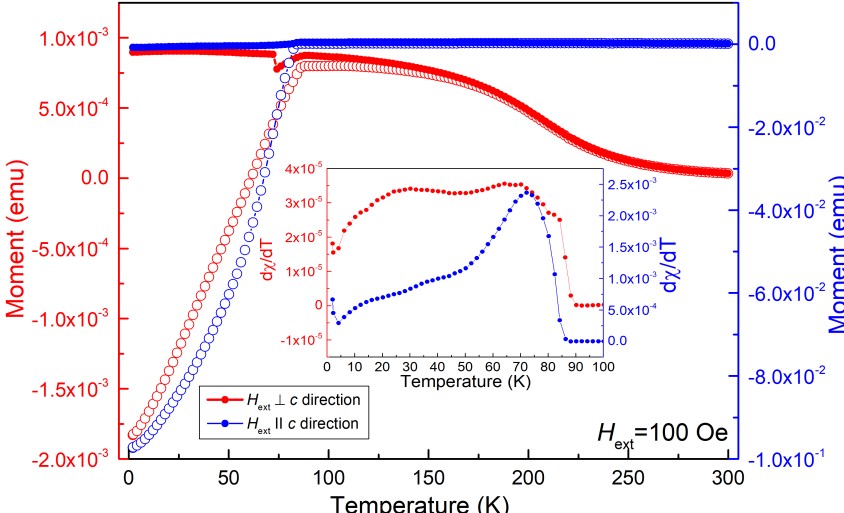

Figure 2: Magnetization ZFC (open circle)-FC (solid circle) data under an applied field of 100 Oe along in-plane and out-of-plane direction to the plane of substrate, inset: first derivative of ZFC susceptibility for both the magnetization direction showing onset of superconducting transition.

The dependence of the onset temperature, $T_{c,SC}$ and the magnetic behavior on the magnetization direction suggests possible coupling of the superconducting layer of YBCO through the ferromagnetic layer of LSMO. Similar observation in bilayer thin films of $La_{0.66}Ca_{0.33}MnO_3$(LCMO)/ YBCO in earlier studies [16] [17] is consistent with these results. It appears that the ferromagnetic ordering in the LSMO layer induces magnetic ordering of the Cu moments in the YBCO film leading to significant renormalization of the magnetism of the LSMO film. The formation of non-collinear magnetic ordering at the interface can persuade spin-triplet component of the superconductivity ordering that has a long-ranged proximity effect in the ferromagnetic layers [18] [19].

In order to examine the effect of external magnetic field in different directions on the superconductivity (low temperature region) and ferromagnetic ordering (higher temperature region) of the heterostructure, magnetic hysteresis loops were measured in both the regions along both the directions. In Figs. 3(a) and 3(b), we show the magnetic hysteresis loops at 20 K (superconducting region) and 100 K (ferromagnetic region) respectively. $M - H$ loop at 20 K shows the characteristic superconducting like hysteresis loop in both the directions. The experimental data of 20 K clearly show that the $\Delta$M value for out-of-plane direction is an order higher than for in-plane direction. Here, $\Delta$M is defined as the difference between the lower and upper branches of magnetic hysteresis loop at a particular field.

Magnetic hysteresis loop at 100 K shows ferromagnetic behaviour and no significant change along the two directions confirming the presence of magnetic anisotropy only in the superconducting region because of the presence of superconducting $CuO_2$ plane along $c$-direction [20].

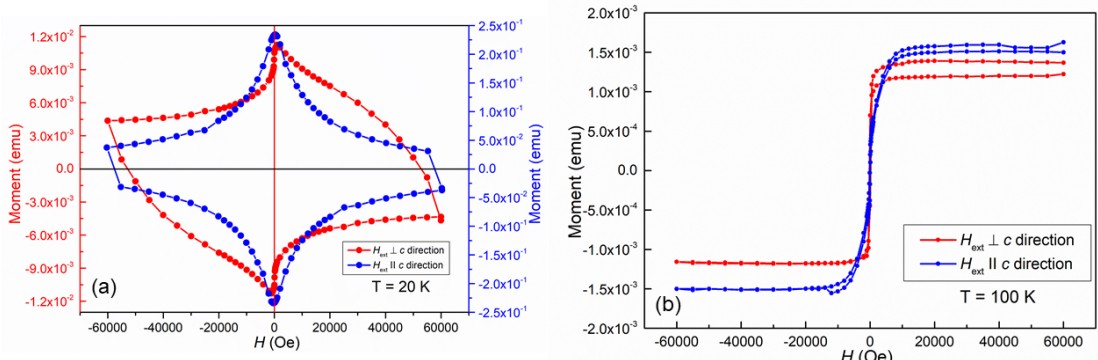

Figure 3: (a) Magnetic hysteresis loop at 20 K showing superconducting behaviour, (b) Magnetic hysteresis loop at 100 K showing ferromagnetic ordering due to the presence of LSMO layer.

## 4 Conclusion

In summary, we synthesized YBCO/LSMO bilayer thin film on STO substrate using pulsed laser deposition and investigated the presence of magnetic anisotropy by applying the magnetic field along two directions i.e. in-plane and out-of-plane direction of the film. X-ray diffraction confirms the orientation of both the layers i.e. LSMO and YBCO along (00$l$) direction indicating uniaxial growth of the film along $c$-direction. Magnetic measurements show the onset temperature for superconductivity, $T_{c,SC}$ around 84 K and 86 K for out-of-plane and in-plane direction respectively. Magnetic hysteresis loop shows an order higher magnetization value for out-of-plane direction as compare to in-plane direction in the superconducting region at 20 K. No change in the magnetization value is observed for hysteresis loop in ferromagnetic region at 100 K confirming the presence of magnetic anisotropy due to the superconducting layer at low temperature. These results suggest that these LSMO/YBCO heterostructures are ideal playground for the study of interplay between magnetism and superconductivity that might help to unravel the underlying physics of unconventional superconductivity.

## Acknowledgements

The authors acknowledge financial support from DAE, Govt. of India (Project No. 12-R&D-TFR-5.10-0100). A. S. acknowledges Infosys-TIFR Leading Edge Travel Grant for the travel fund. K. M. acknowledges financial support from DAE-BRNS, Govt. of India under the DAE-SRC-OI Award program.

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
