# Peer review of "Structural and magnetic anisotropy in YBa2Cu3O7/La0.67Sr0.33MnO3 bilayer on SrTiO3 substrate"

_SciPost Physics Proceedings, doi:SciPost Phys. Proc. 11, 007 (2023)_

## Round 1 · Referee Report · Anonymous (Referee 1) · 2022-9-14

Strengths

  1. Excellent sample quality, which is always tricky to achieve considering retaining single crystalline phase across the heterostructure is always a challenge.

  2. Results from X-ray diffraction studies is clear and definitive.

  3. Clear representation of the results presented.

Weaknesses

  1. Major criticism is the magnetic anisotropy interpretation. It is based on change of the Tc value from H perpendicular to in-plane with c-axis. This is estimated from magnetization studies. But the variation is within 2K (86K to 84K) and in most cases the error bar (deltaT) on exact estimation of Tc value from ZFC curves is comparable to the 2K range. So an inset of first-order derivative showing exact location of peak at Tc would be more convincing and add value to the manuscript.

  2. A discussion on the effect of superconductivity on the magnetism is not addressed adequately. Few possible publications from Prof. C. Bernhard (pioneers in this field) discussing interplay of superconductivity and magnetism in these heterostructure can be used for discussions : Nature Materials 8, pages 315–319 (2009), Physical Review B 87, 115105 (2013), Physical Review B 100, 115129 (2019)

Report

Overall the manuscript reads well. With the recommendations suggested it is suitable for publication.

Requested changes

Mentioned within weakness section.

  • validity: good
  • significance: ok
  • originality: ok
  • clarity: good
  • formatting: good
  • grammar: good

Author:  Ankita Singh  on 2022-10-04  [id 2877]

(in reply to Report 1 on 2022-09-14)
Category:
correction

First, we would like to thank the referee for careful analysis, detailed response, and positive attitude regarding our manuscript. His/her technical comments have helped us substantially to improve our manuscript. We have taken into account his/her comments in revising the manuscript.

  1. We have added the first order derivative of susceptibility curve as inset in Fig.2, clearly showing the change in the Tc value with a difference of 2 K along both the direction of applied magnetic field.

  2. We agree completely with the referee regarding the discussion about the effect of magnetism on superconductivity that was missing in our manuscript. We have added a paragraph on page no. 4 in our manuscript regarding the change in superconductivity/ferromagnetism in bilayer thin films due to the proximity effect.

We again thank the referee for these minor but important comment

Attachment:

Manuscript_revised_Q4QX1wD.pdf

---

## Round 2 · Author Response

We thank both referees for their time in critically evaluating the manuscript, comments, and recommendations. We have revised the manuscript considering all the comments. Response to the comments is given below.
1. Major criticism is the magnetic anisotropy interpretation. It is based on change of the Tc value from H perpendicular to in-plane with c-axis. This is estimated from magnetization studies. But the variation is within 2K (86K to 84K) and in most cases the error bar (deltaT) on exact estimation of Tc value from ZFC curves is comparable to the 2K range. So an inset of first-order derivative showing exact location of peak at Tc would be more convincing and add value to the manuscript.

Reply: We have added the first-order derivative of the susceptibility curve as an inset in Fig.2, which clearly show the change in the Tc, SC with the change in the direction of the applied magnetic field.

2. A discussion on the effect of superconductivity on the magnetism is not addressed adequately. Few possible publications from Prof. C. Bernhard (pioneers in this field) discussing interplay of superconductivity and magnetism in these heterostructure can be used for discussions: Nature Materials 8, pages 315–319 (2009), Physical Review B 87, 115105 (2013), Physical Review B 100, 115129 (2019)

Reply: We agree with the referee and added a paragraph on page no. 4 in the revised manuscript discussing the change in superconductivity/ferromagnetism in bilayer thin films due to the proximity effect. We have also added the suggested references.

---

## Round 2 · List of Changes

1. Page 3, Fig. 2: Added an inset showing the derivative of susceptibility for different magnetization directions. The following text is added to describe the inset.

“Such dependence of the onset temperature for superconductivity, Tc on the direction of externally applied magnetic field is confirmed from the first derivative of the susceptibility (see inset of Fig. 2).”

2. Page 4: Added the following paragraph

“The dependence of the onset temperature, Tc, SC and the magnetic behavior on the magnetization direction suggests possible coupling of the superconducting layer of YBCO through the ferromagnetic layer of LSMO. Similar observation in bilayer thin films of La0.66Ca0.33MnO3 (LCMO)/YBCO in earlier studies [16,17] is consistent with these results. It appears that the ferromagnetic ordering in the LSMO layer induces magnetic ordering of the Cu moments in the YBCO film leading to significant renormalization of the magnetism of the LSMO film. The formation of non-collinear magnetic ordering at the interface can persuade spin-triplet component of the superconductivity ordering that has a long-ranged proximity effect in the ferromagnetic layers [18,19].”

3. Page 4: Added new references [16], [17], [18], and [19].

---

## Editorial Decision

published